# The Role of Ritual in Children's Acquisition of Supernatural Beliefs

**Anna Mathiassen** [1] **and Mark Nielsen** [1,2,*]

1   Early Cognitive Development Centre, School of Psychology, University of Queensland, St. Lucia 4072, Australia; mathiassen.anna@gmail.com

2   Faculty of Humanities, University of Johannesburg, Auckland Park, Johannesburg 2092, South Africa

*   Correspondence: nielsen@psy.uq.edu.au; Tel.: +61-7-3365-6805

**Abstract:** This study investigated how observing the ritualisation of objects can influence children's encoding and defence of supernatural beliefs. Specifically, we investigated if ritualising objects leads children to believe those objects might be magical, buffering against favouring contrary evidence. Seventy-nine children, aged between 3 and 6 years, were presented with two identical objects (e.g., two colour-changing stress balls) and tasked with identifying which was magical after being informed that one had special properties (e.g., could make wishes come true). In a Ritual condition, an adult acted on one of the objects using causally irrelevant actions and on the other using functional actions. In an Instrumental condition, both objects were acted on with functional actions. The children were given a normative rule relating to the use of the objects and an opportunity to imitate the actions performed on them. A second adult then challenged their magical belief. Ritualistic actions increased the likelihood of children attributing magical powers to the associated object but did not affect resistance to change or adherence to normative rules. However, children who engaged in ritual actions protested more when the magical belief was challenged. Our findings suggest that rituals can play an important role in shaping children's perception and defence of supernatural beliefs.

**Keywords:** ritual; supernatural; imitation; belief; magic; development; protest





## 1. Introduction

Most of the world's cultures feature some form of collective religion. Two prominent features constituting religious systems are adherence to some supernatural belief system and engagement in various rituals (Whitehouse 2021). However, the psychological mechanisms that underpin ritual behaviour and how these mechanisms support the contemplation of supernatural entities remain empirically underexplored. We aimed to help bridge this gap in our understanding by investigating the potential link between young children observing ritual actions and their acquisition of supernatural beliefs and how ritualising objects may imbue them with supernatural properties and act as a prophylactic against abandoning belief in the face of contrary evidence. It is important to explore this in young children, given (1) that they are still developing their belief systems and have a more malleable worldview than adults; and (2) the potential implications across the lifespan from early religious and spiritual experiences (Abo-Zena and Midgette 2019).

Supernatural beliefs have been defined as "the attribution of an event to supernatural processes, such as the actions of a supernatural agent (for example, gods, ancestor spirits, human magical practitioners such as witches or shamans) or supernatural force (for example, karma, evil eye)" (Jackson et al. 2023). Such beliefs often contain shared features, such as concepts of a natural observer, creator, judge, or alternative realities (Szocik and Oviedo 2018). The word *supernatural* indicates that these beliefs are outside the natural or observable realm, deeming them non-scientific. Therefore, a key feature of established supernatural beliefs is their resistance to contradictory evidence (Davis 2017). Another

prominent component of supernatural beliefs is their association with rituals. According to Whitehouse (2011), ritualistic actions comprise normative behaviour with an irretrievably opaque causal structure. This definition highlights a ritual's causal opaqueness and goal demotion.

Causal opaqueness refers to actions that do not have a logical link to the desired outcome, such as knocking on wood to avoid bad luck (McGuigan et al. 2007). In defining ritualistic action as comprising inherently causally opaque structures, we highlight that the performed actions are not functionally linked with the outcome. Goal demotion, on the other hand, refers to the observer's ignorance of the motivations or intentions behind specific actions (Nielsen et al. 2018). For example, a cleaning function may be apparent, but it is unclear why a specific cloth is used or why the action is performed while singing a specific chant. These features of ritual behaviour cue individuals to approach the actions differently. That is, they adopt what is known as a *ritual stance* and eschew what is known as an *instrumental stance* (Kapitány and Nielsen 2015).

Several studies have explored children's sensitivity to, and interpretation of, ritualistic actions. For example, Nielsen et al. (2018) presented 3–6-year-old children from Bushman communities in South Africa with a puzzle box in which an adult opened using actions with varying degrees of goal demotion ranging from goal-directed (a sticker was retrieved from the box as a clear outcome), through partial goal demotion (the sticker was retrieved but returned to the box or the sticker remained untouched) to full goal demotion (the box was empty). The children consistently replicated the irrelevant actions across conditions, but those who experienced full goal demotion reproduced these actions at significantly higher rates. As goal demotion is interpreted as a signal of ritual behaviour, these findings highlight how goal demotion can work together with causal opacity to increase children's sensitivity to ritualistic actions and the perceived normative importance placed on them.

Somewhat contrarily, Kapitány et al. (2018a) presented children from communities in Australia and Vanuatu with an adult who enacted ritualised actions on one bowl and non-ritualised actions on another. Two identical stickers were then placed in the bowls, and children were allowed to choose which stickers to take. Regardless of cultural heritage, the ritualisation process did not impact the children's choices, with them being equally likely to choose the sticker from the ritualised as the non-ritualised bowl. However, the target objects were not ritualised in this instance, just the receptacles containing them. Thus, it remains unclear how children would react when given the choice of engaging with an object that has been directly subjected to ritual actions or one that has not.

It has also been documented that children will protest when social or moral norms are breached, even those newly established (Josephs et al. 2016). For example, Rakoczy et al. (2008) presented 2- to 3-year-old children with a novel game in which a demonstrator presented an action labelled with a new pseudo-word ("I am going to show you how to "*dax*""). After the child learned the action associated with daxing, a puppet performed an incorrect action while either claiming to dax or that it was showing the child something new. The children protested much more when the puppet claimed to be daxing. This finding suggests that children can quickly establish normative beliefs to the degree that they are willing to protest a breach only minutes after the introduction of a novel concept. However, it is unclear if these results transfer from a game scenario to rituals and the establishment of supernatural beliefs.

In combination, the afore-discussed studies highlight that children are sensitive to ritual actions as normative information and are willing to defend even newly established normative beliefs. However, the extent to which ritualising objects sways children's conceptualisation of them as supernatural and functions to imbue them with resistance to alternative interpretations remains largely untested. Our aim in the current study was to begin bridging this gap in the literature.

To experimentally investigate the role of rituals in encoding supernatural beliefs, we explored an analogue—a novel magical belief—rather than relying on pre-established supernatural systems. There is a foundation for this in that positive associations have

been found between the number of ritualised compulsion-like behaviours a child performs and their inclination toward accepting magical beliefs (Evans et al. 2002). Through this approach, we explored how low arousal rituals alone potentially assist the acquisition of magical beliefs. Specifically, we expected that children would be more likely to allocate magical powers to an object if the object was paired with a ritualistic action sequence (goal demoted and causally opaque) than an instrumental action sequence (goal-directed and causally transparent). Additionally, we expected ritualising to strengthen the belief in the object's magical powers, making the belief more resistant to change and the children more likely to protest in the face of contradictory evidence. All specific hypotheses were pre-registered and can be found here (OSF, https://osf.io/r34c7).

## 2. Results

### 2.1. Preliminary Analyses

Some gender differences in imitation rates have been previously reported (Frick et al. 2017; Schleihauf et al. 2019). In contrast, the current study failed to reveal any differences between males and females regarding actions directed to the target and the non-target object (see Table 1). Similarly, bivariate correlations between gender and the other target variables showed no significant association $p > 0.05$ (See Appendix C for bivariate correlations). As there were no a priori justifications for other effects, we did not consider gender further.

**Table 1.** Mean imitation rates (and standard deviations) by gender and associated T-tests.

|  | Males | | Females | | | | | | |
|---|---|---|---|---|---|---|---|---|---|
|  | *M* | *SD* | *M* | *SD* | *t* | *df* | *p* | *d* | **95% CI** |
| Target Object | 2.4 | 3.4 | 2.4 | 2.4 | −0.03 | 77 | 0.973 | −0.01 | −1.08, 1.05 |
| Non-target Object | 1.5 | 1.6 | 1.4 | 1.8 | 0.29 | 77 | 0.775 | 0.07 | −0.66, 0.89 |

*Note*: CI = confidence intervals.

Consistent with previous literature (Speidel et al. 2021), we found a positive correlation between age and imitation for the target object—$r = 0.26$, $p = 0.023$—such that older children imitated at higher rates compared to younger children. However, this correlation was not significant for the non-target object—$r = 0.21$, $p = 0.06$. In addition, there was a significant correlation between age and unprompted copying of I1—$r = 0.22$, $p = 0.050$—such that as age increased, children were more likely to imitate the ritual actions with higher fidelity and more likely to copy I1 unprompted. No other variables were associated with age, so age was not considered further.

All children passed the manipulation check by reciting back the rule. As only three participants copied the actions demonstrated by I1 without being prompted to do so, the variable of *unprompted copying of I1* was removed from subsequent analyses.

Assumption Checks

Assumption checks were conducted to ensure that the statistical analyses were appropriate. Kolmogorov–Smirnov tests revealed that most Dependent Variables (DVs) breached the normality assumption for the parametric tests (See Appendix D). Furthermore, Protest for Disputed Magic and Protest for Breach of Rule fell outside the generally accepted cut-off value for Skewness: ±2.0 (2.26, 3.50), and Kurtosis: ±7.0 (7.42, 13.59, respectively). However, according to Knief and Forstmeier (2021), parametric t-tests are robust to violations of normality and are more powerful than non-parametric tests. Therefore, both parametric and non-parametric tests were conducted to account for breaches of assumptions and the limited range of scores. The results of the analyses were consistent across tests, and the significance of the findings did not differ between the two methods (see Appendix E for non-parametric results). Thus, only parametric tests are reported below, in line with pre-registration.

### 2.2. Hypothesis Testing

The pre-registered hypotheses explored whether children in the Ritual and Instrumental conditions differed in their Perception of Magic, Level of Protest, the Fidelity of Copying and Spontaneous Actions. We collapsed the children's congruent scores across trials for all DVs to increase the robustness of the results.

#### 2.2.1. H1: Perception of Magic

In line with pre-registered hypothesis H1.1, children were more likely to infer magical properties to the target object in the Ritual condition ($M = 1.34$, $SD = 0.73$) compared to the Instrumental condition ($M = 0.87$, $SD = 0.63$),—$t(77) = 3.09$, $p = 0.003$, $d = 0.70$, and 95% CI [0.17, 0.78], as identified by our dichotomous measure of Perception of Magic at T1 collapsed across trials, providing a single score per participant ranging from 2 (the target object is identified as the most magical on both trials) to 0 (the non-target object was identified as magical on both trials). Contrary to hypothesis H1.2, there was no significant difference between the Ritual condition ($M = 0.90$, $SD = 0.80$) and the Instrumental condition ($M = 0.66$, $SD = 0.78$) in whether or not children retained their beliefs after being exposed to contradictory evidence,—$t(77) = 1.37$, $p = 0.174$, $d = 0.31$, and CI [−0.11, 0.60], as identified by our dichotomous measure of Perception of Magic at T2 collapsed across trials, providing a single score per participant ranging from 2 (identification consistent with T1 on both trials) to 0 (identification contrary to T1 on both trials). However, most children (35) changed their minds on both trials, and a further 26 did on 1 trial (see Table 2). Notably, while the contradictory evidence swayed most children, one-fifth of those tested were not. Furthermore, contrary to H1.3, there was no significant difference between the conditions in the decline in the children's rating of magic, as identified by the Likert scales, for the target and non-target objects (see Table 3).

**Table 2.** Perception of Magic at T2 (after contradictory evidence) frequency and percentage.

| Condition | Ritual | Instrumental | | |
|---|---|---|---|---|
| | *n* | *n* | N | Total % |
| Consistent with T1 on both trials | 15 | 20 | 35 | 44.3% |
| Consistent with T1 on one trial | 15 | 11 | 26 | 32.9% |
| Contrary to T1 on both trials | 11 | 7 | 18 | 22.8% |
| Total | 41 | 38 | 79 | 100% |

Note: Each child went through two trials; one involving percussion eggs and one involving stress balls. The child's assigned condition (ritual, instrumental) stayed consistent throughout the trials.

**Table 3.** The decline in Ratings of Magic from T1 to T2.

| | Ritual Condition | | Instrumental Condition | | | | | | |
|---|---|---|---|---|---|---|---|---|---|
| | *M* | *SD* | *M* | *SD* | *t* | *df* | *p* | *d* | 95% CI |
| Target Object | −0.1 | 2.8 | −0.5 | 2.8 | 0.34 | 77 | 0.738 | 0.08 | −1.21, 1.70 |
| Non-target Object | −0.5 | 2.9 | 0.5 | 2.9 | 0.25 | 77 | 0.805 | −0.06 | −1.57, 1.22 |

*Note*: T-tests failed to reveal a significant difference between conditions in how children's ratings of magic declined between T1 and T2, as identified by a 5-point Likert Scale, 1 = not magical at all, 5 = very, very magical.

#### 2.2.2. H2: Protest

Contrary to the pre-registered hypothesis, there was no significant difference between the Ritual condition ($M = 1.3$, $SD = 2.7$) and the Instrumental condition ($M = 1.1$, $SD = 2.2$) in the amount of protest children expressed when the experimenter violated the normative rule—$t(77) = 0.39$, $p = 0.696$, $d = 0.09$, and 95% CI [−0.88, 1.31]. However, there was a significant difference between conditions in the protest children expressed when the magical belief was disputed—$t(77) = 2.59$, $p = 0.011$, $d = 0.58$, and CI [0.18, 1.40]—such that children in the Ritual condition ($M = 1.3$, $SD = 1.6$) protested at a significantly higher rate compared to the Instrumental condition ($M = 0.5$, $SD = 1.0$).

### 2.2.3. H3: Fidelity of Copying

In line with hypothesis H3.1 and previous research, we found that children imitated actions at a much higher rate for the target object in the Ritual condition ($M = 3.4$, $SD = 2.5$) than in the Instrumental condition ($M = 1.4$, $SD = 1.7$)—$t(77) = 4.13$, $p < 0.001$, $d = 0.93$, and 95% CI [1.03, 2.96]. However, there was no significant difference between the number of actions imitated for the non-target object between the Ritual condition ($M = 1.8$, $SD = 1.8$) and the Instrumental condition ($M = 1.2$, $SD = 1.6$)—$t(77) = 1.56$, $p = 0.122$, $d = 0.35$, and CI [−0.16, 1.36]. Contrary to H3.2, there was no correlation between the fidelity of copying of the target object and the children's resistance to change after contradictory evidence—$r = 0.01$, $p = 0.937$—such that the children's beliefs were no more resistant to contradictory evidence if they engaged in higher fidelity copying of the ritual actions.

### 2.2.4. H4: Spontaneous Actions

Here we were interested in the potential differences between conditions in the children's likelihood to breach the normative rule and their preferential engagement with the target or non-target object. Our results revealed no significant difference between the Ritual condition ($M = 1.32$, $SD = 1.57$) and the Instrumental condition ($M = 1.82$, $SD = 1.37$) for the children's likelihood to breach the normative rule—$t(77) = −1.50$, $p = 0.139$, $d = −0.34$, and 95% CI [−1.16, 0.16]. Similarly, there was no significant difference between the Ritual condition ($M = 0.44$, $SD = 0.74$) and the Instrumental condition ($M = 0.50$, $SD = 0.83$) in the children's preferential engagement with either of the objects—$t(77) = −0.34$, $p = 0.731$, $d = −0.08$, and CI [−0.41, 0.29].

## 3. Discussion

The vast majority of the world's children develop in environments with considerable exposure to supernatural beliefs. Alongside their need to learn how to use the objects and artefacts surrounding them, they must also learn which of these supernatural beliefs they should attend to and which ones they should adopt. Here we aimed to explore the impact of ritualising objects on children's acquisition of supernatural beliefs by documenting whether observing a simple ritualistic action sequence instead of a matched instrumental sequence would increase the likelihood of children adopting a novel magical belief and impact their reaction to subsequent contradictory evidence. This endeavour affords insight into the ways certain beliefs can be formed early in life. This is important given research highlighting the impact of parentally induced religious beliefs on children's social, emotional, and academic development (Bartkowski et al. 2019), and how early established beliefs can be sustained into adulthood (Myers 1996). It also allows insight into the core aspects of human nature. As Machluf and Bjorklund (2015, p. 27) wrote: " . . . the origins of humans' social nature and cognition are found in infancy and childhood, placing social cognitive development at center stage in understanding the evolution of the human mind".

Consistent with the hypotheses, observing an adult treat an object with ritual actions increased the children's belief that the object had magical properties. Contrary to predictions, pairing a magical belief with a ritual did not make it more resistant to change in the face of contradictory evidence. Additionally, the children were no more likely to protest a rule violation when we paired the rule with a ritual action sequence than an instrumental action sequence. Interestingly, the children protested more when the magical property of a ritualised object was disputed compared to a non-ritualised object. Further, the children imitated ritual actions with higher fidelity than instrumental actions, but higher fidelity imitation did not impact their beliefs' resistance to change in the face of contradictory evidence. Finally, we explored whether rituals might be related to the children's willingness to comply with normative rules, their preferential engagement with objects, and their likelihood of copying the informant without being prompted. However, no associations were found.

In line with previous research and our primary hypothesis, our data revealed that the children ascribed magical properties more consistently to ritualised objects than non-

ritualised objects. Notably, the children attributed magical properties to the objects themselves without being explicitly told which object was magical. These findings suggest a potential causal link between engaging in ritualistic actions and building supernatural beliefs, supporting theoretical frameworks that outline rituals as powerful tools for spreading religious doctrine (Whitehouse and McQuinn 2013). Notably, observing and engaging in ritualistic actions did not impact the initial magical belief's resistance to change in the face of contradictory evidence. Rather, we found that the children would weigh the validity of conflicting adult testimony and adjust their beliefs accordingly. However, this lack of support for our hypothesis may be due to the children's pre-existing knowledge about the objects' functions, which could have hindered their acceptance of them as magical. Future research should pilot test various appropriate objects to assess those with low familiarity and potentially more effective in eliciting a magical belief. It is also notable that in this experiment, neither adult was aligned with the children in any way. Thus, this calls for an empirical investigation charting whether or not ritualising objects reinforces the sustenance of magical beliefs when the model is a clear in-group member.

Contrary to the hypothesis, no differences were detected in the likelihood of the children protesting normative rule violations when engaging in ritual actions compared to instrumental actions. A potential explanation for this result is that the first informant presented the normative rule as equally valid for the ritualised and the non-ritualised object. Future research could explore if specifying that the rule only applies to the magical object increases children's inclination to protest. Interestingly, the children were significantly more likely to protest a new adult disputing the magical properties of an object in the Ritual condition compared to the Instrumental condition. Thus, the increased level of protest in the Ritual condition compared to the Instrumental condition indicated that the children are more invested in a belief encoded through ritual actions. This somewhat contrasts with our finding that observing a ritual did not make the belief more resistant to contradictory evidence. These inconsistencies highlight the importance of considering the nuances of how beliefs are learned and reinforced in children. Further research may explore these nuances by varying the type and quality of contradictory evidence, which has previously been found to impact how rationally children are to revise prior beliefs (Schleihauf et al. 2022). This could provide a greater understanding of the degree to which rituals may strengthen children's investment in the belief and whether the ritual itself serves as a form of cognitive priming that reinforces the belief.

Consistent with our hypothesis and existing literature (Fong et al. 2021), the children tended to replicate ritual actions with higher fidelity than instrumental ones. This adds to the growing body of research suggesting that children take a ritual stance that prioritises the process of performing actions over their outcomes (Kapitány and Nielsen 2015). Furthermore, the retention of causally opaque ritual actions is more accurate than instrumental actions, indicating that children are more likely to remember irrelevant actions than functional ones (Kapitány et al. 2018b). Our study suggests that participants pay more attention to and remember actions when presented with a ritual action sequence than a functional one. Surprisingly, we found that the rate of imitation did not correlate with the strength of children's beliefs, despite previous research suggesting a positive association between the two (Evans et al. 2002). Our findings indicated that the repetition frequency within a simple, low-arousal ritual is insufficient to strengthen a supernatural belief. One possible explanation for this inconsistency with previous research is our study's lack of repetition over time.

While not the primary focus of our study, we also examined spontaneous actions, including breach of normality, preferential engagement, and unprompted copying. We failed to find significant differences in the children's likelihood of violating the normative rule between conditions. This suggests that engaging in a ritual action does not prevent children from breaching a novel normative rule presented to them by an unfamiliar adult. We speculate that this may be due to the rule not being sufficiently solidified in the children's minds or that the rule was not specified as being tied directly to the magical object.

In contrast to previous research with adults (Kapitány and Nielsen 2015), we did not find evidence that the children preferred the object presented with ritual actions over the object presented with instrumental actions. This inconsistency suggests that a preference for ritualised objects may develop later in life, a perspective somewhat aligned with prior research by Kapitány et al. (2018a), who found that children were no more likely to choose stickers from a ritualised bowl than from a non-ritualised bowl. However, in contrast to the current study, the target objects were not ritualised, just the bowl containing them. Alternatively, our study's lack of evidence of a preference for ritualised objects could be due to a lack of sensitivity in our measure. We did not explicitly ask the children about their object preferences, leading to few children expressing any preference. New research is needed to investigate this preference more explicitly by asking children what object they prefer. Testing a more comprehensive age range of children could provide insight into when such preferences start to develop.

This study provides valuable insights into the role of rituals in encoding supernatural beliefs. However, several limitations should be addressed in future research. One is the need for more information on the children's religious or cultural backgrounds. Children who have already developed supernatural thinking appear more susceptible to accepting religious and supernatural concepts as real (Davoodi et al. 2016). Therefore, controlling for previous exposure to religious and supernatural beliefs in future studies could provide a more nuanced understanding of the relationship between rituals and supernatural thinking. Another is the potential for the revealed function of the objects used in the experiment, such as the colour-changing balls, to generate excessive excitement and overshadow the rule, reducing the likelihood of protest. Future research could explore the effect of the arousal level of alternative perspectives on children's ability to revise their beliefs by employing objects with varying degrees of excitement. Lastly, as already noted, prior research has shown that beliefs may grow stronger when challenged, depending on the initial belief's strength and the quality of reasoning for the contradictory evidence (Lord et al. 1979; Schleihauf et al. 2022). Therefore, investigating how the quality of reasoning for contradictory views could moderate children's protest and the likelihood of retaining a novel magical belief in future studies would be an exciting avenue for further research.

The current study used a novel magical belief as a proxy for supernatural or religious doctrine, and uncovered insight suggesting that rituals likely play a role in forming and strengthening supernatural beliefs. Our results indicate that when a belief is paired with ritualised actions, children more readily accept and more vigorously defend the belief if subsequently challenged. It also suggests that violating normative rules within a belief system may be less confronting than disputing the belief itself. However, the results of the current study should be considered preliminary, as our study is the first to employ this novel design. Nevertheless, rituals can be secular or religious, and they can be communal or personal. Using the paradigm introduced here presents a promising avenue for charting the ways these various belief system perturbations become acquired and perpetuated.

Acquiring supernatural beliefs in childhood is a complex process that relies on the interplay between many factors. Children from both religious and secular backgrounds may have seen ritualistic actions being performed before in conjunction with supernatural doctrine. Reflecting the ways children tend to understand ritualistic actions as more normative than functional, they may simply accept the objects as magical because of previous exposure from family, community members, and even media. Additionally, the current study only investigated simple, low-arousal rituals performed over a short period. Although researchers have found that low-arousal rituals have been used as a tool for spreading religious doctrine, we know that repetition over time is an important aspect of doctrinal rituals (Whitehouse and McQuinn 2013). Further, research has found that high-arousal rituals need less repetition, create stronger bonds and more outgroup hostility than low-arousal rituals (Whitehouse and McQuinn 2013). As very few studies have tested the specific psychological mechanisms of how ritualistic actions are linked to the development of supernatural beliefs, further research is needed to detangle the details of several factors

that may be involved. Therefore, a flexible research design that can vary isolated factors is now needed to capture the complexity of how rituals contribute to the acquisition of supernatural beliefs in childhood.

Our study does, though, offer a novel approach to experimentally investigate the interplay between various ritual actions, normative behaviour, and magical thinking. This design provides a strong foundation for further exploration of the acquisition of supernatural and religious beliefs. Although our study replaced supernatural and religious beliefs with magic, the same psychological principles of appealing to something transcending the natural world are likely to apply. Our findings can, thus, shed light on these specific aspects of religious belief systems. Moreover, the current study has important implications for gaining insight into the complexities of human cognition, cultural transmission, socialisation processes, and emotional development. It can help us understand how children's minds construct and interpret their environment, including concepts of the supernatural. Rituals play a fundamental role in cultural transmission (Legare and Nielsen 2015). They contribute to the sharing of beliefs, traditions, and values through generations. Investigating how rituals influence children's supernatural beliefs can shed light on the mechanisms of cultural transmission and the interplay between culture and cognition. Understanding how these beliefs are acquired and perpetuated can have implications for preserving cultural heritage and fostering intergenerational understanding.

Furthermore, exploring how rituals influence the emotional reactions to challenging established beliefs can help us understand the emotional and psychological functions these rituals serve and their potential impact on children's well-being and emotional regulation. By better understanding the complexity of rituals on the development of supernatural beliefs, educators and researchers can develop more effective strategies for children to distinguish between healthy and unhealthy supernatural thinking. With this knowledge, educators may be better equipped to challenge unhealthy beliefs while simultaneously understanding the social, emotional, and cultural value of supernatural beliefs. Such knowledge can inform culturally sensitive education and intervention programs to promote critical thinking skills and foster social cohesion across belief systems. Future research could build upon our design to examine how the complex interplay between factors—such as the strength of reasoning, type of ritual, and the level of excitement associated with opposing perspectives—moderate children's evaluation of supernatural information. Such an investigation could further illuminate the interplay between rituals and the development of supernatural beliefs and shed light on some of the core features of the human condition.

## 4. Method

### 4.1. Participants

The study recruited 87 children between 3 and 6 years of age from the database of a child development centre that is part of a large metropolitan university. Children of this age range were selected as it is a period in which sensitivity to ritualised actions and to conventional versus instrumental verbal cues becomes established (Moraru et al. 2016; Nielsen et al. 2015). The gender breakdown by age is presented in Table 4. Of the participants, 54 were identified by their guardians as Caucasian, 13 as Asian, 6 as Latino, 4 as African, and 2 from other backgrounds. The guardians of 63 children (80%) reported completing a university degree or higher. Eight children were excluded: four due to experimenter error, three due to the child's inability to finish the experiment, and one because of guardian interference. Participants were split between 2 conditions (41 in the experimental condition and 38 in the control condition). While previous studies of ritual with children included 16–20 participants per condition (Nielsen et al. 2015; Wilks et al. 2016), our aim here was to include more due to the novelty of the design while being mindful of the potential for an underpowered analysis (Brysbaert 2019).

The study employed a between-group design, with children randomly assigned to one of two conditions. Each child participated in two trials, as described below.

**Table 4.** Summary of the final sample of participants by age and gender.

|        | 3 Years | 4 Years | 5 Years | 6 Years | Total |
|--------|---------|---------|---------|---------|-------|
| Female | 6       | 11      | 11      | 13      | 41    |
| Male   | 8       | 11      | 9       | 10      | 38    |
| Total  | 14      | 22      | 20      | 23      | 79    |

*4.2. Materials*

The study utilised two sets of objects. One trial involved two stress balls that changed colour from blue to pink when squeezed, and the other trial used two black percussion eggs that made a rattling sound when shaken. The objects were displayed on stands (see Table 5). A white tissue was used as a cleaning agent in the Egg trial. In both trials, the study used a laminated five-point size-and-numbers-based Likert scale (see Table 5).

**Table 5.** Table of study materials, including test material and Likert scale.

| | **Egg Trial** | **Ball Trial** |
|---|---|---|
| Test Material | 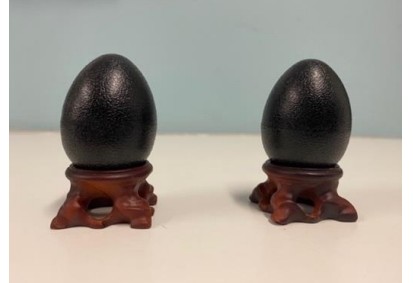  Percussion Eggs | Stress Balls |

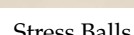

| | |
|---|---|
| Scale | Likert Scale Used to Indicate Perception of Magic |

*Note*: Likert scale: 1 = Not magical at all; 5 = Very, very magical.

*4.3. Procedure*

All task administration was conducted at child-friendly test facilities. Written consent was obtained from the guardians of each child prior to participation. Upon arrival, children participated in a warm-up activity intended to familiarise them with the experimenter and the two informants and to create a space where they felt comfortable expressing their opinions even if they contradicted the adult present. The activity was derived from Rakoczy et al. (2008) and involved the facilitating experimenter wrongfully naming familiar toys and waiting to be corrected by the child. For example, the experimenter picked up a toy cow and asked the child if they "liked this horse". This activity lasted five to ten minutes until the child appeared comfortable expressing their opinions. Each child was then escorted to a testing room and positioned across from the facilitating experimenter. Guardians who wished to observe the session were seated behind the child's back to ensure minimal influence on their behaviour.

4.3.1. Testing Session

The first set of test objects (two percussion eggs or two stress balls) was displayed on the table, and the experimenter explained that one was magical but did not identify

which one (See Appendix A for the full script and Appendix B for more detail of the test procedure). The child was asked to help figure out which object was magical (e.g., for the percussion eggs: *I know that one of these eggs is magical. Actually, one of them can make all your wishes come true. But I am not sure which one it is, so I need your help to figure it out*) and the experimenter exited the room. Informant 1 (I1) entered to perform condition-dependent action sequences on each object, introducing a normative rule relating to both objects (e.g., for the percussion eggs: "we must never shake them"). For a visual representation of the action sequences, see Table 6. I1 then exited, and the facilitating experimenter re-entered.

**Table 6.** Visual representation of the ritual and instrumental action sequences.

| | Egg Trial | | Ball Trial | |
| --- | --- | --- | --- | --- |
| | Ritual [a] | Instrumental [b] | Ritual [c] | Instrumental [d] |

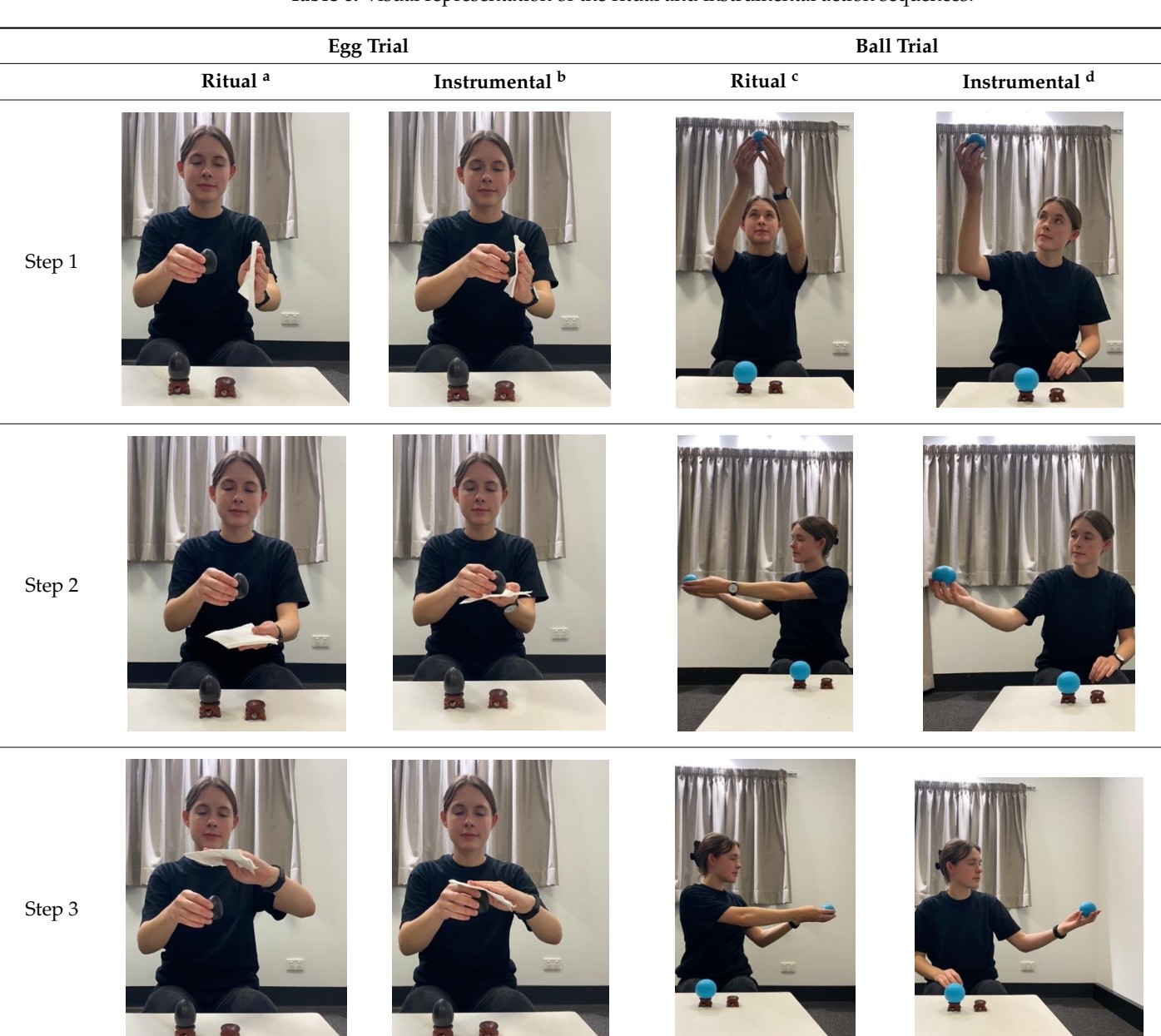

[a] Informant moved her hands slowly in a wiping motion as if cleaning the percussion instrument from the side, below, and above, respectively, but without touching the tissue to the object; [b] Informant cleaned the percussion instrument using a wiping motion in patterns that matched the Ritual condition, but touching the tissue to the object; [c] Informant lifted the stress ball with both hands in a slow and rigid fashion over her head, to her right, and to her left, respectively, while looking at the object reverentially; [d] Informant lifted the stress ball with one hand in a neutral fashion with matched patterns to the Ritual condition while looking at the ball to "inspect it for dirt spots". The images here depict the experimenter administering the task. Permission for these images to appear has been granted by the experimenter.

The experimenter asked the child to perform the modelled actions, allowing them to relay their newly acquired information verbally and physically, which has been shown to strengthen beliefs (Richert et al. 2022). The experimenter then asked the child to identify which of the objects was magical and rate the degree of perceived magic associated with each item using the size-based Likert scale. The child was also asked to recall the normative rule before the experimenter left the room.

The experimenter exited as Informant 2 (I2) entered the room and disputed the magical properties of the object the child previously identified as being the most magical, revealing the contradictory functional purpose of the objects and indicating that they were going to break the normative rule (e.g., *"they are musical instruments, you shake them to make music. Let me show you when I shake it"*). I2 paused for a couple of seconds, allowing time for the child to protest, before continuing the actions for three seconds regardless of the protest, and then left the room.

The experimenter re-entered and asked the child again to identify the most magical object and each object's respective degree of magic using the Likert scale. The child was then given 15 s alone with the objects before the experimenter returned and ended the trial. Each child participated in two trials, with the trials differing in the presented object (percussion egg/stress ball), paired action sequences (cleaning/looking for spots), and proposed magical properties (" . . . *can make your wishes come true/can turn bad things into good things"*). The testing session took approximately ten minutes, and the sessions were videotaped for coding purposes with informed consent from the guardian. Upon completion of both trials, the guardian was debriefed, and the child received a certificate and a small gift for participation.

### 4.3.2. Conditions

Children were randomly assigned to one of two conditions: Ritual or Instrumental. Both conditions were conducted identically except for the presented action sequences. In the Ritual condition, one object was paired with a ritualistic action sequence involving causally opaque and goal-demoted actions, while the other object was paired with a closely matched instrumental action sequence (see Table 6). In the Instrumental condition, both objects were paired with instrumental actions. A random number generator was used to ensure full counterbalancing of the items first used, the side of the item first picked up (left or right), the side of the target item, and the gender of I1. To be clear: all children were presented with the egg and the stress ball demonstrations. Children in the Ritual condition experienced these demonstrations with ritualised actions included, whereas children in the Instrumental condition did not.

### 4.3.3. Perception of Magic

At T1, the children were asked to identify the magical object. For both conditions, children were coded as having chosen the non-target object (coded as 0) or the target object (coded as 1). In the Ritual condition, the target object was specified as the ritualised item. As both objects were subject to the functional action sequence in the Instrumental condition, counterbalancing determined which object was designated as the target object for coding purposes to facilitate comparison. At T2, we coded whether the children selected the same object as they did at T1 (coded as 1) or the alternate object (coded as 0). Furthermore, the degree of perceived magic associated with each object was measured using the five-point Likert scale at both T1 and T2 (1 = not magical at all; 5 = very magical).

### 4.3.4. Copying

Copying was recorded for both ritualistic and instrumental actions in three ways. First, we noted if the child engaged with the same object as I1 (coded as 0 = no; 1 = yes). Second, we coded whether the child attempted to reproduce the actions performed on the object (coded as 0 = no; 1 = yes). Any cleaning or looking for dirt was considered copying for both objects. Finally, we counted the number of actions imitated out of three possible actions for

each object, regardless of the condition (coded as 0–3). Imitation was coded based on the object's movement, and no imitation was counted if the object was not lifted off the stand. If the actions could not be separated, only one action was counted.

### 4.3.5. T2 Protest

Following Rakoczy et al. (2008), the protest was classified as 0 = no protest, 1 = hints of protest, and 2 = explicit protest. Hints of protest were recorded if the children frowned, moved their mouths as if they were about to speak, or moved their arms as if they were about to stop I2 from breaching the normative rule. Because of the subtle nature of hints of protest, a child would get a score of 1 if any hints were present, regardless of the number of hints displayed. The explicit protest was recorded if the child verbally protested or physically stopped C2 from breaking the normative rule. All explicit protest was counted such that if the child protested several times, the sum of the protests was recorded. However, the explicit protest was counted as multiple protests only if the child paused between phrases and/or actions. The protest was coded separately for reactions to *contradictory evidence of magic* and *breach of normative rule* with the same scoring criteria, depending on when the protest occurred.

### 4.3.6. Spontaneous Actions

Spontaneous actions were recorded for (1) breach of normative rule (coded as 0 = no; 1 = yes when left alone at the end of each trial; 2 = yes throughout the experiment), (2) imitating action sequences without prompt (coded as 0 = no; 1 = yes when left alone at the end of each trial; 2 = throughout the experiment), and (3) preferential engagement with the object (coded as 0 = no unprompted engagement; 1 = target object preference; 2 = non-target object preference; 3 = equal object engagement).

**Author Contributions:** Conceptualization, A.M. and M.N.; methodology, A.M. and M.N.; formal analysis, A.M.; investigation, A.M.; resources, M.N.; data curation, A.M.; writing—original draft preparation, A.M.; writing—review and editing, M.N.; funding acquisition, M.N. All authors have read and agreed to the published version of the manuscript.

**Funding:** This study was funded by an Australian Research Council Discovery Project Grant to Mark Nielsen and Andrew Whiten (DP140101410).

**Institutional Review Board Statement:** This study meets the requirements of the National Statement on Ethical Conduct in Human Research (2007, current revision) and was approved by the University of Queensland Health and Behavioural Sciences Low and Negligible Risk Ethics Board (2022/HE000626; 19 April 2022).

**Informed Consent Statement:** Informed consent was obtained from the legal guardian of all subjects involved in the study.

**Data Availability Statement:** Raw data is available from the authors on request.

**Acknowledgments:** We thank Jane Minogue and Domenic Randall for their assistance in data collection and all the families who generously gave their time to participate.

**Conflicts of Interest:** The authors declare no conflict of interest.

## Appendix A.

*Appendix A.1. Script*

Appendix A.1.1. Ball Trial

The experimenter (E) brings the child into a room where two stress balls (ball 1 & ball 2) are displayed.

EXPERIMENTER (E)

*Look at what we have here; Do you know what? I know that one of these is actually magical. Do you want to know what one of them can do? One of them can make our wishes come true. That's pretty cool, right? But the problem is I don't know which one. So, I'm going to go get Person 1, and we can see what he/she does to them. Then I need your help to figure out which one of these is magical. Do you think you can help me with that?*

E leaves, and Person 1 (P1) enters.

RITUAL CONDITION:

PERSON 1

*Oh, I know what these are. It is very important that we treat them correctly and that they are clean. I will show you how we treat this one.*

P1 performs the ritualistic actions on Ball 1 (Lifting the ball reverentially, looking at it from both above and beneath, causally opaque and goal demoted) ORDER OF RITUAL AND INSTRUMENTAL ACTIONS ARE COUNTERBALANCED.

PERSON 1

*I will show you how we check this one for dirt.*

P1 performs the non-ritualistic actions on Ball 2 (similar movements to the ritualistic actions only with the clear goal of inspecting the ball, causally transparent and goal-directed)

INSTRUMENTAL CONDITION:

PERSON 1

*Oh, I know what these are. It is very important that we treat them correctly and that they are clean. I will show you how we check them for dirt.*

P1 performs the non-ritualistic actions on both balls (See ritual condition).

BALL TRIAL CONTINUES . . .

PERSON 1

*But, you know, we should never squeeze them. Now, you can have a go.*

Pushes the balls over, allowing the child to perform the actions.

PERSON 1

*Well done. Now, do you remember what we shouldn't do?* (Reminds the child if they indicate they don't know). *Actually, I have to go now, but I will get (E) for you. I will be back later and show you some other things (only in the first trial)*

P1 leaves the room, and E enters.

EXPERIMENTER

*Hi, how did you go? So, what did you learn about these? Do you what to show me what P1 did with them?*

Pushes the balls over and watches the child perform the actions.

EXPERIMENTER

*Well done. So now I'm curious. Which one of these do you think is magical?*

Allows the child point to one of the objects.

EXPERIMENTER

*Awesome. Can you tell me how magical you think it is by pointing somewhere on this scale for me? If you think it's very, very magical, you point to the big circle, and if you think it's not magical at all, you point to the little circle, or you can point anywhere in between.*

Holds out the circle Likert scale and waits for the child to decide.

EXPERIMENTER

*Awesome. So, you think this one is the magical one (points to the one they chose), so this one, probably not so much then, huh? (Point to the other one) So how magical do you think this one is?*

Holds out the Likert scale and lets the child pick a number.

EXPERIMENTER

*Ok, thank you. Do you remember what we must never do with these?* (Reminds them of the rule if they forget: *"we must never . . . . squeeze them"*) *Great! I also know that P2 wants to see you, so I am going to go get him/her and then I'll be right back.*

E leaves. Person 2 (P2) enters and looks at the stress balls.

PERSON 2

*I think know what these are. I don't think this one is magical* (pointing to the ball the child has identified as magical) *I think they are stress balls, and if you squeeze them, they change colour. Let me show you when I squeeze this one* (pointing to the one the child identified as magical).

Pauses slightly before picking up the stress ball the child had identified as magical, squeezing it for three seconds, and then returning it to its stand.

PERSON 2

*I am going to go get (experimenter's name) for you. Bye.*

P2 leaves and E enters.

EXPERIMENTER

*Ok, I am back. We are just about done with these, but before we finish up, I just wanted to ask you again; Which one of these do you think is the magical one now?*

Letting the child point to one of the objects

EXPERIMENTER

*Can you please show me how magical you think it is? Remember, this means "very, very magical", this means "not magical at all", but you can also point to any of these in between.*

Holds out the Likert scale for the child to indicate the level of magic.

EXPERIMENTER

*And what about this one?*

Directed at the other object, holds out the Likert scale for the child to indicate the level of magic.

EXPERIMENTER

*Thank you for playing with us. You can have a look at them now if you want to while I go get some other things for you to look at/while I pack up.*

The child is left alone for 15 s to see what they do with the objects before (E) re-enters with the objects needed for trial 2 (only in the first trial).

Appendix A.1.2. Egg Trial

The experimenter (E) brings the child into where two percussion eggs (Egg 1 and Egg 2) are displayed.

EXPERIMENTER (E)

*Look at what we have here; Do you know what? I know one of these is actually magical. Do you want to know what one of them can do? One of them can turn bad things into good things. That's pretty cool, right? But the problem is I don't know which one. So, I am going to get X, and we can see what she/he does to them. Then I need your help to figure out which one of these is magical. Do you think you can help me with that?*

E leaves, and Person 1 (P1) enters.

RITUAL CONDITION:

PERSON 1

*Oh, I know what these are. It is very important that we treat them correctly and that they are clean. I will show you how we treat this one.*

P1 performs the ritualistic actions on Egg 1 (cleaning process without touching the egg, causally opaque and goal demoted).

ORDER OF RITUAL AND INSTRUMENTAL ACTIONS ARE COUNTERBALANCED (see Coding Sheet).

PERSON 1

*I will show you how we clean this one.*

P1 performs the non-ritualistic actions on Egg 2 (a cleaning process with similar movements to the ritualistic actions, only actually touching the egg, causally transparent and goal-directed)

INSTRUMENTAL CONDITION:

PERSON 1

*Oh, I know what these are. It is very important that we treat them correctly and that they are clean. I will show you how we clean them.*

P1 performs the non-ritualistic actions on both Egg 1 and Egg 2 (see ritual condition)

EGG TRIAL CONTINUES . . .

PERSON 1

*But, you know, we should never shake them. Now, you can have a go.*

Pushes the eggs over, allowing the child to perform the actions.

PERSON 1

*Well done, and do you remember what we shouldn't do? (*Reminds the child if they don't know*). Actually, I have to go now, but I will get (E) for you. I will be back later and show you some other things (only in the first trial).*

P1 leaves the room, and E enters.

EXPERIMENTER

*Hi, how did you go? What did you learn about these? Do you want to show me what he/she did with them?*

Pushes the eggs over and watches the child perform the actions.

EXPERIMENTER

*Well done. So now I'm curious. Which one of these do you think is magical?*

Let's the child point to one of the objects.

EXPERIMENTER

*Awesome. Can you tell me how magical you think it is by pointing somewhere on this scale for me? If you think the ball is very, very magical, you point to the big circle and if you think it's not magical at all, you point to the little circle you can point anywhere in between.*

Holds out the circle Likert scale and waits for the child to decide.

EXPERIMENTER

*Awesome. So, you think this one is the magical one (*points to the one they chose*), so this one, probably not so much the, huh? (*Points to the other one*) So how magical do you think this one is?*

Holds out the Likert scale and lets the child pick a number.

EXPERIMENTER

*Ok, thank you. Do you remember what we must never do with these?* (Reminds them of the rule, if they forget: *"We must never . . . shake them"*)

*Great! I also know that X wants to see you, so I am going to go get her/him and then I'll be right back.*

E leaves. Person 2 (P2) enters and looks at the eggs.

PERSON 2

*Oh, I think I know what these are. I don't think this one is magical (*pointing to the egg the child had identified as magical*). I think they are musical instruments, and if you shake them, they make music. I'll show you when I shake this one (*pointing to the egg the child has identified as magical*).*

Pauses slightly before picking up the percussion egg, shaking it for three seconds, and then placing it down in the same spot.

PERSON 2

*I am going to go get (experimenter's name) for you. Bye.*

(P2) leaves, and (E) enters.

EXPERIMENTER

*Ok, I am back. We are just about done with these, but before we finish up, I just wanted to ask you again; Which one of these do you think is the magical one now?*

Letting the child point to one of the objects.

EXPERIMENTER

*Can you please show me how magical you think it is? Remember, this means "very, very magical, this means "not magical at all", but you can also point to any of these in between.*

Holds out the Likert scale for the child to indicate the level of magic this object holds.

EXPERIMENTER

*And what about this one?*

Directed at the other object, the experimenter holds out the Likert scale for the child to indicate the level of magic.

EXPERIMENTER

*Thank you for playing with us. You can have a look at them now if you want to, while I go get some other things for you to have a look at/while I pack up.*

The child is left alone for 15 s to see what they do with the objects before (E) re-enters with the objects needed for trial 2 (only in the first trial).

**Appendix B.**

*Summary of Task Administration Procedure*

| Procedure Step | Facilitator | Measure | Ritual Condition | Instrumental Condition |
|---|---|---|---|---|
| *Warmup exercise* | *Experimenter, Informant 1, Informant 2* | *Child's confidence in expressing opinions that are contradictory to adult statements* | *In a dedicated play area, the experimenter and the informants introduce themselves to the child and their guardian while engaging in child-directed play with their preferred toys. The experimenter wrongfully names specific toys to encourage the child to correct the experimenter and subsequently praises the child for the correction.* | |
| Introduction to Objects | Experimenter | none | Two identical percussion eggs are introduced | |
| Introduction of Actions | Informant 1 | none | I1 performs ritual actions on one egg and instrumental actions on the second egg | I1 performs instrumental actions on both eggs |
| Introduction of Normative Rule | Informant 1 | none | The child is told never to shake the eggs, a rule equally directed at both objects. | |
| Imitation | Informant 1 | Rates of Copying | The child is provided with the opportunity to copy the actions performed by I1 | |
| T1—Evaluation | Experimenter | Magic T1, Degree of Magic T1 | The child is asked to point to the most magical object and rate the degree of perceived magic associated with each item using a visual Likert scale. | |
| Introduction of Contradictory Evidence/Violation of Rule | Informant 2 | Protest Disputed Magic, Protest Rule Violation | The child is told the object they identified as the most magical at T1 is not magical. The object's function is revealed, and the normative rule is violated. | |
| T2—Evaluation | Experimenter | Magic T2, Degree of Magic T2 | The child is again asked to point to the most magical object and rate the degree of perceived magic associated with each item using the same visual Likert scale. | |
| Exploration | Experimenter | Unprompted Copying, Preferential Engagement, Rule Violation | The experimenter leaves the child alone with the objects for 30 s for individual exploration. | |

Note: Each child went through two trials (counterbalanced) such that all children experienced both sets of objects (percussion eggs and stress balls). The condition stayed consistent through trials.

## Appendix C.

*Preliminary Bivariate Correlations*

Bivariate correlations for gender, age, and imitation

| Correlations | | Gender | Age in Months | Imitation Number Ritual | Imitation Number Instrumental |
|---|---|---|---|---|---|
| Gender | Pearson Correlation | 1 | 0.069 | 0.004 | −0.033 |
| | Sig. (2-tailed) | | 0.543 | 0.973 | 0.775 |
| | N | 79 | 79 | 79 | 79 |
| Age In Months | Pearson Correlation | 0.069 | 1 | 0.256 * | 0.214 |
| | Sig. (2-tailed) | 0.543 | | 0.023 | 0.058 |
| | N | 79 | 79 | 79 | 79 |
| Imitation Number Ritual | Pearson Correlation | 0.004 | 0.256 * | 1 | 0.682 ** |
| | Sig. (2-tailed) | 0.973 | 0.023 | | 0.000 |
| | N | 79 | 79 | 79 | 79 |
| Imitation Number Instrumental | Pearson Correlation | −0.033 | 0.214 | 0.682 ** | 1 |
| | Sig. (2-tailed) | 0.775 | 0.058 | 0.000 | |
| | N | 79 | 79 | 79 | 79 |

*. Correlation is significant at the 0.05 level (2-tailed). **. Correlation is significant at the 0.01 level (2-tailed).

## Appendix D.

*Test of Normality*

| Tests of Normality | Condition | Kolmogorov-Smirnov [a] | | |
|---|---|---|---|---|
| | | Statistic | df | Sig. |
| Gender | Instrumental | 0.352 | 36 | **0.000** |
| | Ritual | 0.357 | 41 | **0.000** |
| Age In Months | Instrumental | 0.142 | 36 | 0.064 |
| | Ritual | 0.114 | 41 | 0.200 * |
| Perception of Magic at Time1 (Which one of these do you think is magical?) | Instrumental | 0.308 | 36 | **0.000** |
| | Ritual | 0.305 | 41 | **0.000** |
| Perception of Magic at Time2 (Which one of these do you think is magical?) | Instrumental | 0.327 | 36 | **0.000** |
| | Ritual | 0.236 | 41 | **0.000** |
| Change Ratings for Ritual Object (Rating at T1 minus T2) | Instrumental | 0.118 | 36 | 0.200 * |
| | Ritual | 0.246 | 41 | **0.000** |
| Change Rating for Instrumental Object (Rating at T1 minus T2) | Instrumental | 0.180 | 36 | **0.005** |
| | Ritual | 0.198 | 41 | **0.000** |
| Protest—disputed Magic | Instrumental | 0.442 | 36 | **0.000** |
| | Ritual | 0.232 | 41 | **0.000** |
| Protest—breach of Rule | Instrumental | 0.353 | 36 | **0.000** |
| | Ritual | 0.318 | 41 | **0.000** |
| Imitation for Ritual Object | Instrumental | 0.302 | 36 | **0.000** |
| | Ritual | 0.256 | 41 | **0.000** |
| Imitation for Instrumental Object | Instrumental | 0.301 | 36 | **0.000** |
| | Ritual | 0.231 | 41 | **0.000** |
| Normality Breach | Instrumental | 0.181 | 36 | **0.004** |
| | Ritual | 0.263 | 41 | **0.000** |
| Preferential Engagement | Instrumental | 0.401 | 36 | **0.000** |
| | Ritual | 0.430 | 41 | **0.000** |
| Unprompted Copying | Instrumental | 0.534 | 36 | **0.000** |
| | Ritual | 0.538 | 41 | **0.000** |

*. This is a lower bound of the true significance. [a] Lilliefors Significance Correction. Note: All dependent variables are collapsed across trials.

## Appendix E.

*Non-Parametric Tests*

| | Null Hypothesis | Test | Sig.[a,b] | Decision |
|---|---|---|---|---|
| Hypothesis Test Summary of Non-Parametric Tests | | | | |
| H1.1 | The distribution of Perception of Magic at Time 1 is the same across categories of Condition. | Independent-Samples Mann-Whitney U Test | 0.003 | Reject the null hypothesis. |
| H1.2 | The distribution of Perception of Magic at Time 2 is the same across categories of Condition. | Independent-Samples Mann-Whitney U Test | 0.161 | Retain the null hypothesis. |
| H1.3a | The distribution of Change Rating for Ritual Object (T1-T2) is the same across categories of Condition. | Independent-Samples Mann-Whitney U Test | 0.496 | Retain the null hypothesis. |
| H1.3b | The distribution of Change Rating for the Instrumental Object (T1-T2) is the same across categories of Condition. | Independent-Samples Mann-Whitney U Test | 0.760 | Retain the null hypothesis. |
| H2x | The distribution of Protest—Disputed Magic is the same across categories of Condition. | Independent-Samples Mann-Whitney U Test | 0.003 | Reject the null hypothesis. |
| H2 | The distribution of Protest—Breach of Rule is the same across categories of Condition. | Independent-Samples Mann-Whitney U Test | 0.457 | Retain the null hypothesis. |
| H3.1a | The distribution of Imitation for Ritual Object is the same across categories of Condition. | Independent-Samples Mann-Whitney U Test | 0.000 | Reject the null hypothesis. |
| H3.1b | The distribution of Imitation for Instrumental object is the same across categories of Condition. | Independent-Samples Mann-Whitney U Test | 0.117 | Retain the null hypothesis. |
| H4.1 | The distribution of Preferential Engagement is the same across categories of Condition. | Independent-Samples Mann-Whitney U Test | 0.799 | Retain the null hypothesis. |

[a] The significance level is 0.050. [b] Asymptotic significance is displayed.

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
