# Peer review of "The Role of Ritual in Children’s Acquisition of Supernatural Beliefs"

_religions, doi:10.3390/rel14060797_

Round 1

Reviewer 1 Report

This paper contributes to research on supernatural belief formation among young children. I pose the following questions to sharpen the arguments.

1. Please make a stronger case for significance. Early childhood is a foundational period of life and it is possible that beliefs formed then will be carried into later life. The authors nod in this direction but could be more convincing.

2. The age range of the children studies (3-6 years old) is a broad spectrum in terms of cognitive development. Provide assurance that the age criterion was appropriate. If other published studies have used this age range, be sure to indicate that.

3. Britannica is not a legitimate citation, and there is no corresponding reference provided. Rely on scholarly sources for definitions of key terms.

4. There is considerable evidence that parents exercise a great deal of influence in children's early religious development (see work by Bartkowski on religion and child development). How does this research fit into what's investigated and discovered in the current paper?

5. I am not an expert on quantitative methods. Is there sufficient power in the sample for this investigation?

6. The study identifies how rituals of a specific sort influence supernatural belief formation, but there are different types of rituals, some religious and some less so (communal worship, private prayer, magical thinking, etc.). Please give additional consideration to this issue in the Discussion. Also, what about supernatural belief formation in an organizational context (congregation) versus outside of it? Could the discussion speak to those issues as well?

Subject-verb agreement and a few other issues are evident. A careful proofreading by a professional English copy editor is recommended.

Reviewer 2 Report

This a very good paper that I enjoyed reading, however before publishing kindly go through the following points

1. The procedure that you adopted needs some clarification. A table would help the reader. 

2. Line 101 You also need to explain the substantial (not actual) difference between the control group and the experimental group since all children conducted the experiment either with the black eggs or with the stress ball - with both objects having a ritual or a nonritual attached to them.

3. Clarify how Informants 1 and 2 were introduced to children

4. Re: Analysis - On line 273 you make reference to M = 1.34. Am I understanding correctly this refers to M on the Likert Scale Used 1-5 to indicate Perception of Magic? If this is the case children were, in general not very convinced with the magical properties of the object. Kindly clarify.

5. In this regard Table 6 does not help to clarify the matter above. If in the experimental design, there truly is a difference between the control group and the experimental group then this needs to be shown in the table. Furthermore, it would benefit the readers if the table is presented in more detail, ie detailing Frequency and Percentage for T1 and T2 for both black eggs and stress balls.

6. It would be beneficial if you would show if there was any statistical difference between the age groups. Were younger children more susceptible to identifying and sticking with a magical object?

7. Line 434. Do the results really point to the 'key' role of rituals in the acquisition of supernatural belief? If anything they support the possibility that they might have a role. In your research design, it could be argued that the Informants were actually telling a story mainly through nonverbals. Children are obviously not tabula rasa and, even if they are not part of a religious community they still have their prior knowledge and prior experiences which they also acquire through the media. Rightly so you said that another research would need to control for children's religious background and doctrine but one also needs to take into consideration other cultural or societal factors. For instance, it is not unusual for 4-year-old children who are either secular or whose family practices a Christian religion to adopt facial expressions and use their hands in a manner that reminds them of Buddhist practices when introduced to mindful activities (particularly using silence). This is most probably due to the influence that social media has on children in a globalized society. I, therefore, invite you to acknowledge the complexity and the many variables that are at interplay)

8. Your comment in lines 438 and 442  is not clear - in what manner would the implications for education and intervention programs foster critical thinking and foster social cohesion? By convincing and removing beliefs in the supernatural?

9. In the reference list include 'H.' after the last references t the work of Whitehouse

Round 2

Reviewer 1 Report

I commend the authors on a sound revision. 

1. Please address statistical power in relation to sample size directly. If it’s been done, I missed it. 

The authors could restate the number of subjects per condition for their study as compared with previously published investigations.

2. The Bartkowski and Bartkowski citation should be  Bartkowski, Xu, and Bartkowski. 

I have no concerns re language. 

Author Response

We appreciate the continued constructive feedback regarding our manuscript. It has certainly improved it.

Regarding power, we state in the Participants sections:

Participants were split between two conditions (41 in the experimental condition and 38 in the control condition). While previous studies of ritual with children included 16-20 participants per condition (Nielsen et al., 2015; Wilks et al., 2016), our aim here was to include more due to the novelty of the design while being mindful of the potential for underpowered analysis (Brysbaert, 2019). 

Contemporary developmental psychology studies of this nature typically have cell sizes around 20, some extending to 30. Given ours are upwards of this number we continue to believe that we have sufficient power to detect medium to large effects. As we provide effect sizes for all of our primary results any scholar wishing to test for power, either to question our findings or as a basis for continued work, will be able to do so.

Thanks for picking up the errant citation to Bartkowski, Xu, and Bartkowski. We have now fixed this in the latest version.